# Effective Reversal of Macrophage Polarization by Inhibitory Combinations Predicted by a Boolean Protein–Protein Interaction Model

**DOI:** 10.3390/biology12030376

**Published:** 2023-02-27

**Authors:** Gabor Szegvari, David Dora, Zoltan Lohinai

**Affiliations:** 1Translational Medicine Institute, Semmelweis University, 1094 Budapest, Hungary; 2Department of Anatomy, Histology and Embryology, Semmelweis University, 1094 Budapest, Hungary; 3Pulmonary Hospital Torokbalint, 2045 Torokbalint, Hungary

**Keywords:** macrophage polarization, protein–protein interaction network, Boolean control network, tumor microenvironment, combination therapy, drug target selection

## Abstract

**Simple Summary:**

Understanding the effects of the tumor microenvironment is an essential step to advance treatments for cancer, but it is also one that is problematic to reproduce in vitro or study in vivo. When such approaches are difficult to implement, in silico methods are often able to help surmount these barriers. Network models of biological systems focus on the interactions of components (such as proteins or genes) and use available data about the parts of a system to predict emergent qualities. We focused our study on macrophages and how their environment affects their polarization. Thus, we built a Boolean control network model of the early response events going on in macrophages by collating information from a manual search of the literature that interprets the changes in the inner state of the cell. We used this model to simulate combinatorial treatment options that target multiple targets at the same time and have synergistic effects on macrophage polarization.

**Abstract:**

Background: The function and polarization of macrophages has a significant impact on the outcome of many diseases. Targeting tumor-associated macrophages (TAMs) is among the greatest challenges to solve because of the low in vitro reproducibility of the heterogeneous tumor microenvironment (TME). To create a more comprehensive model and to understand the inner workings of the macrophage and its dependence on extracellular signals driving polarization, we propose an in silico approach. Methods: A Boolean control network was built based on systematic manual curation of the scientific literature to model the early response events of macrophages by connecting extracellular signals (input) with gene transcription (output). The network consists of 106 nodes, classified as 9 input, 75 inner and 22 output nodes, that are connected by 217 edges. The direction and polarity of edges were manually verified and only included in the model if the literature plainly supported these parameters. Single or combinatory inhibitions were simulated mimicking therapeutic interventions, and output patterns were analyzed to interpret changes in polarization and cell function. Results: We show that inhibiting a single target is inadequate to modify an established polarization, and that in combination therapy, inhibiting numerous targets with individually small effects is frequently required. Our findings show the importance of JAK1, JAK3 and STAT6, and to a lesser extent STK4, Sp1 and Tyk2, in establishing an M1-like pro-inflammatory polarization, and NFAT5 in creating an anti-inflammatory M2-like phenotype. Conclusions: Here, we demonstrate a protein–protein interaction (PPI) network modeling the intracellular signalization driving macrophage polarization, offering the possibility of therapeutic repolarization and demonstrating evidence for multi-target methods.

## 1. Introduction

The immune system has to react to a variety of threats in a localized and targeted manner. To accomplish this, immune cells go through various changes that define their function in response to external stimuli. As part of this process, identical groups of T lymphocytes and macrophages can either drive regenerative, wound healing events or trigger inflammation and an aggressive response to detrimental agents. Disruption of this finely regulated equilibrium plays an essential role in several pathologies, causing excessive inflammation [1,2,3,4] or hampering immune surveillance and activation, which can be responsible for cancer formation [5,6]. Immune cells, and in particular, tumor-associated macrophages (TAMs), are essential components of the tumor microenvironment (TME) [7,8]. Still, even though their therapeutic potential is gaining traction [9,10,11,12,13], both in vitro assays and experimentation on murine models face significant hurdles in accurately reproducing the TME and macrophage function [14,15,16].

Macrophages go through a process called polarization when exposed to certain stimuli, giving rise to a broad spectrum of effector cells [17]. This functional change is supported, and to an extent defined, by an extensive change in the cell’s transcriptome [18,19]. To avoid being targeted by the immune system, cancerous cells often develop an ability to influence these changes by modulating the TME [20]. By expressing and secreting immunomodulating agents, they can drive a shift in function from tumoricidal to regenerative, tissue-protective, and even tumorigenic responses in TAMs [21,22]. M1-polarized macrophages represent a tumoricidal phenotype and secrete reactive oxygen species (ROS) and cytokines, including IL-1 and TNF-α, associated with microbicidal and pro-inflammatory activities [23]. M2-polarized macrophages express the CD163 antigen, are regulated by IL-4 and -13, and are described as tumorigenic [24,25].

Drug and therapy design has always been an arduous task: partly due to the daunting amount of options for possible targets and drug molecules; partly due to technical issues inherent in creating physiologically adequate circumstances in vitro or interpreting the results of in vivo tests. While we still cannot wholly replace these steps, computationally based in silico technologies can expedite them and make systematic searches practically feasible. Developing a mathematical model of a system allows us to create predictions for a host of scenarios and combinations of circumstances [26,27]. To derive meaningful information about perturbations on the cellular level, the model needs to incorporate proteins and genes. The direct biochemical modeling of intracellular events is beyond the current state of computing, but modeling the interactions of these elements and the flow of information through their network has proven to be a successful approach [28]. Based on publicly available data, network models of different complexities can be constructed. Reducing the information stored about individual components to a binary (on/off) state and building what is known as a Boolean network might seem deceptively simple, but it still allows for meaningful deductions about the system, while also enabling the use of diverse resources to build it [29]. This reduction in the number of parameters included also allows for the examination of more complex scenarios, especially a systematic study of multi-target therapies or drugs [30].

In the current work, we present a detailed in silico model of the intracellular processes leading to the polarization of macrophages in response to their environment, relying primarily on direct interactions between components and with a particular focus on the TME. The system is stimulated with physiologically relevant external stimuli and its polarization status is inferred from changes in the transcriptome of the macrophage. Exploring perturbations that represent therapeutic action, we show the effect of single and combinatory inhibitions of crucial protein targets in the signal transduction of macrophage polarization in a presumptive TME.

## 2. Materials and Methods

### 2.1. Network

In this model, the intracellular signaling pathways of a macrophage are represented with a Boolean control network (BCN) with a synchronous update scheme. The system receives input in the form of small molecule transmitters from extracellular space, and the transcriptional pattern of selected genes relevant to polarization is considered the output. The activation status of all elements (represented as nodes) is registered, and the spread of signals via their interactions (represented as edges) is tracked from receptors to transcription factors and genes (Figure 1).

We focused on the early response of the cell, meaning that effects that require de novo protein synthesis (including auto/paracrine effects, e.g., in the case of IFNβ) and feedback loops (e.g., in the case of SOCS3) were excluded. Our model abstracts the passage of time to an extent that these effects would mask the initial response of the cell. Moreover, our aim was to examine the fundamental decision the cell makes about its state. Without a change in the environment, all these later responses are fully dependent on the early response. They either reinforce or hold in check a primary decision already made by the system.

Being a Boolean model, each node has two states: active and inactive. The behavior of the network is evaluated in subsequent steps. In each step, all nodes simultaneously update their state based on the states of the source nodes of all incoming edges and the weight of those edges. The state of input nodes (small molecule transmitters) is fixed in each run and is used to represent the microenvironment of the cell. Steps are taken until an attractor is reached (a stable cycle of network states, possibly consisting of only one state), at which point the state of output nodes is examined. Attractors are recognized with our implementation of Brent’s cycle detection algorithm [31]. The mathematical formulation of model state progression is based on the one presented in [32]. The state of the system is calculated from its state in the previous step as follows:(1)$$S_{t+1}=thr\left(W\cdot S_{t}-\theta \right)$$
where S_x_ is the state of the system at step x, represented as a binary column vector; W is a square weight matrix containing all edge weights (with nonexistent edges having a weight of 0); Θ is a vector of threshold values for each node; and thr is a threshold function defined as: [if x ≥ 0: thr(x) = 1, otherwise thr(x) = 0].

### 2.2. Nodes

We performed a systematic search of research articles describing the effects of extracellular stimuli on macrophage polarization in terms of changes in gene expression. Receptors (to serve as input nodes) and genes (to serve as output nodes) that have multiple available references with unprocessed raw data documenting their impact on macrophage polarization, showing that they are often used to induce it or as markers of it, respectively, were selected as the frame of the model [22,33,34] (see also the references in Appendix A). This was carried out both to restrict the size of the model (compared to extracting an extensive list from a database) and to make it feasible to verify the model based on published research data. In addition, in order to incorporate a gene into the model, information describing the transcription factors necessary and sufficient for its expression was necessary. We are not aware of any large-scale database of this information and manually curated all protein–gene interactions. Potential output nodes without sufficient support were excluded [22]. The nodes connecting the two sides of the model are identified using the KEGG Pathway database [35]. We focused on pathways that we expected to be described in the literature in sufficient detail to make the translation of their constituent processes into our Boolean representation possible. We selected the MAPK, TLR, PI3K/AKT and Jak/STAT pathways, as together they contain all the input and output nodes and establish signal transductory linkages between them. As signaling often relies on combinatorial effects, this core list was expanded with first neighbor nodes that create crosstalk amongst these pathways through the edges found in the interaction databases (see Edges). In addition, in some cases we were unable to satisfy the verification criteria with the nodes identified systematically, and hypothesized that an element missing from the model is responsible for the discrepancy. We performed a manual search of the literature to reveal proteins with considerable impact on these processes that were not included or sufficiently connected in the databases (e.g., STK4 has no outgoing edge in KEGG and no link to any other part of our model in STRING). These were included if by doing so the criterion in question could be satisfied.

Input nodes represent signaling molecules in extracellular space, output nodes represent the transcriptional activation of genes, and most inner nodes represent signal transducer proteins. In addition, some inner nodes represent a group or complex of proteins based on functional considerations, e.g., the components of the PI3K complex are separate proteins, but they function as subunits forming a functional unit and are thus assigned to a single node. In these cases, interactions of the parts were considered to be interactions of the group.

Proteins that are not expressed in monocytes or macrophages were also excluded (e.g., T cell-specific proteins). The expression profile of CD14+ monocytes was checked in the database presented in [36], accessed through biogps.org. Similarly, proteins with constitutive activity in monocytes or macrophages (e.g., the transcription factor PU.1) were also excluded, as their inactivation would likely cause a deviation from the macrophage phenotype and constantly active nodes would hold no information. The complete list of nodes, including classification and expression data, can be found in Appendix A.

### 2.3. Edges

Edges represent functional interactions and the flow of information in the signaling network. Interactions are based on the SignaLink2 [37], STRING [38] (accessed on 26 March 2021) and HPRD [39] databases. Due to efficiency considerations, we filtered the STRING database for physical interactions (tagged as ”binding”). Connections based only on in silico predictions (reasonably common in SignaLink2) or merely on involvement in the same pathway (often referenced in STRING) were not included, meaning that edges connecting genes (i.e., output nodes) to the network are all based on a systematic search of the literature.

All the edges identified in these datasets were then manually curated to determine their direction and sign (i.e., whether they are stimulatory or inhibitory), and they were only included in the model if these parameters were clearly supported by the literature (preferring primary research articles). Edges without direct evidence for their function in human macrophages were added or removed during the verification process if there was substantial evidence in other cell types and the change improved the model’s adherence to the verification criteria.

It should be noted that not all edges represent direct physical interaction (e.g., PI3K that acts through PIP3). Moreover, not all physical interactions are included as edges. MyD88 and TRAF6 bind each other in a larger complex, but the flow of activation goes through IRAK1. While complex formation and scaffold proteins are certainly an essential factor to consider in general, the effects of these binding events are too subtle from the viewpoint of our relatively coarse Boolean system and are, thus, excluded.

The weight of the edges is used to create logic gates to represent the complex ways in which the activation of these entities depends on each other. It is not supposed to represent any measurable interaction parameter, such as binding affinity or reaction speed. For example, if two nodes send edges of weight 0.5 to the same target node, it means that they are both necessary for the activation of the downstream node (AND gate), not that they each affect 50% partial activation by themselves (See Equation (1), above). Inhibitory interactions are represented with a negative weight but otherwise follow the same rules as stimulatory ones. The complete list of edges, including supporting references (PubMed IDs), is presented in Appendix A.

### 2.4. Model Evaluation

At step 0, the system always starts with all nodes inactive. The constitutive activation or inhibition of nodes is introduced by changing their threshold value for activation (in Θ of Equation (1)). Once an attractor is reached, the state of the cell is characterized with a polarization index (pI), following the spectrum model of macrophage polarization [17], which is calculated as:(2)$$pI\left(i\right)=\frac{A_{1}\left(i\right)}{T_{1}}-\frac{A_{2}\left(i\right)}{T_{2}}$$
where A_x_(i) is the number of active outputs of type x for input i and T_x_ is the total number of such outputs. Type 1 is inflammatory (M1-like), and type 2 is tissue-protective (M2-like). For designations, see Appendix A. In some cases, the role of specific genes in polarization is ambiguous; in these cases, we focused on effects in the TME. When evaluating a limit cycle, the ratio of states in which each node is active is summed for the appropriate nodes to derive A_x_(i). In this model, we consider the following outputs to be type 1 (inflammatory, M1-like): CD64, CD68, CIITA, CXCL10, IFNβ, IL-12A, IL-12B, IL-1B, IL-6, iNOS, TNFα; and the following to be type 2 (tissue-protective, M2-like): CCL17, CCL22, CCL24, CD209, MERTK, MRC1, TGFβ.

The synergistic action of two inhibitions is measured with a synergy index (synI), defined as:(3)$$synI\left(A,B,i\right)=\frac{\left|\Delta pI\left(AB,i\right)\right|}{max\left(\left|\Delta pI\left(A,i\right)\right|+\left|\Delta pI\left(B,i\right)\right|,\tau \right)}$$
where ΔpI(X,i) is the change in the polarization index due to the inhibition of node(s) X compared to no inhibition under the same input conditions, i. 𝜏 is an adjustment parameter used to avoid excessive index values due to low ΔpI values for single inhibitions, as these would be misleading. For our calculations we use τ = 0.1, but this is an arbitrary choice. Calculations were performed using MATLAB (The MathWorks, Inc., Natick, MA, USA) and code is available as Appendix A.

## 3. Results

### 3.1. Network Characteristics

Our network consists of 106 nodes, divided into 9 input, 22 output and 75 inner nodes, and 217 edges connecting them (Figure 1). Appendix A show nodes and edges, respectively. Intracellular interaction networks are usually expected to be scale-free [40]. Our model being a control network, its degree distribution is somewhat distorted compared to other biological networks. Nodes with no output or input are only those explicitly designated as inputs or outputs, respectively, with few exceptions for technical reasons (JAK-s are constitutively active with no input but affect the network only via receptor complexes). Thus, nodes with a degree of 1 are relatively underrepresented. Compared to expectations, we also obtained an abundance of nodes with a degree of 4. One factor is that technical nodes are overrepresented in this group (8 out of 11 have a degree of 4), but even excluding them, the number remains above the expected value. If we exclude degrees 1 and 4 as outliers, the network is scale-free, i.e., it shows a power law distribution (R2 = 0.80) (Figure 2a).

Checking the betweenness centrality of nodes (Figure 2b), we can see that proteins recognized as major actors retain this position in our network too, confirming that our selection of nodes did not skew the pathways involved. As expected from our decision to exclude feedback mechanisms, the network attractors are all steady-state with 0 or 1 nodes inhibited. However, with two nodes inhibited, limit cycles do appear in a limited number of cases (2894 or 0.26%). Network components are predominantly localized and are expected to interact in the plasma membrane, cytosol, and the nucleus (Figure 2c).

### 3.2. Verification

While the internal mechanisms of the cell are less frequently explored in detail, numerous studies report changes in gene expression in response to external stimuli in macrophages. We used these data to verify our model. Experimental results linking each of our chosen inputs to our outputs were collected to create verification criteria. We gathered 55 single input-to-output references via a systematic search of the literature (Appendix A). We then compared these data to the output of our model when only that single input node was active. The results were used to expand our literature search, including nodes not initially found in the databases to explain faulty input-to-output connections. Edges representing fairly well-documented relationships but without direct evidence in macrophages were also reviewed to minimize deviations. The presented model satisfies 52 of the 55 verification criteria (94.55% match).

In addition, we examined how well our model fits the general understanding of the function of the inputs and outputs. The polarization index was calculated for inputs in the same way as for outputs (see Materials and Methods, Equation (2)), and we compared the two for all input combinations (Figure 3). Although the correlation between the two is moderate (Pearson r = 0.580), there is a clear tendency, as expected. The disappearance of tissue-protective outputs (pI < −0.2) and the appearance of inflammatory ones (pI > 0.2) happens around the same input pI, even though it is a little below 0. Nevertheless, there is a fairly well defined switch, as input pI increases. While our polarization index is close to continuous, we observed that outputs form three loose groups regarding polarization: an M1-like, an M2-like, and a transitory or uncommitted (Figure 3). This is the basis of setting the thresholds for repolarization at ±0.2 when examining the effects of simulated interventions.

### 3.3. Single Inhibitions Versus Synergistic Combinations

We examined how the disruption of signal transduction would affect the position of our network on the spectrum of polarization, in order to model the effects of small-molecule inhibitors applied as drugs. We ran simulations with one or two of the inner nodes (except receptors) inhibited (immutably set to inactive) for all input combinations. This shed light on some properties of the system.

While there are a number of single node inhibitions that cause a considerable shift in the polarization index (pI) of the final state of the model (Figure 4), these almost never cause a change in polarization or what we refer to here as a “flip”. In more concrete terms, we say that a perturbation causes a flip in the context of an input combination if, both with and without the perturbation, the model converges on an attractor with abs(pI) > 0.2, but the actual values in the two cases are of opposite signs. Of a total of 34,237 cases (67 targets × 511 input combinations), 790(2.31%) cause a ΔpI over +0.4 (a shift towards an inflammatory state) and 2210(6.46%) cause a ΔpI under −0.4 (a shift towards a tissue protective state); however, out of those, only 4 and 0, respectively, cause a flip. Thus, if we wish to reeducate the system, we require a different approach.

The simultaneous inhibition of two nodes is much more effective for these purposes. Of the 1,129,821 cases (2211 target combinations × 511 input combinations), we found 326 that flip towards inflammation and 202 that flip towards tissue protection. While this leaves a fair number of options when looking for therapeutic targets, it is important to note that these comprise only a meager portion of all combinations showing an abs(pI) of > 0.4, 0.79% and 0.16%, respectively. This means that the general trend that flipping is considerably harder than strengthening an existing polarization or steering an unclearly polarized system in one direction still holds, even with combinations of targets.

To examine these in more depth, we calculated a synergy index (synI) that compares the individual contributions of the two inhibited targets to their combined effect (Figure 5; see also Equation (3) in Materials and Methods). A synI of 1 means that the two inhibitions create their effects independently of each other. A synI under 1 usually results from the targets being part of the same pathway, rendering their combination not as effective as could be expected. (However, we have to note that we assume 100% inhibition efficiency.) A synI over 1 means that the combination is more effective than just the sum of its parts. Most of the cases (75.10%) show some overlap in function between the inhibited nodes, and the rest mostly indicate involvement in parallel pathways (23.14%), with only a small portion (1.76%) exhibiting true synergistic effects. Of the 202 that flip towards tissue protection, 200 show a synI over 1. Most are a combination of the inhibition of NFAT5, a target of negligible effect (ΔpI > −0.1) with another target, that in itself would bring the system to a pI around 0. This means that in almost all the cases, a target must be included for which screening for individual targets would not identify as effective. It also reveals that there are converging pathways that need to be blocked to achieve a flip in this direction.

The picture is not this clear in the other direction. Most of the effective combinations (75.5%) involve STAT6, the only node that can flip as a single target. The most notable increase in effectiveness here is from the fact that the combinations can affect a wider range of input combinations. Combining STAT6 with STK4 increases the number of affected inputs to 10 (45%), and pairing with the inhibition of Sp1 raises it to 14 (64%). Combinations not including STAT6 also offer a way to impact the system in a wider variety of environments, with only four (18%) input combinations lacking a target combination that would create a flip. Another notable observation is of a combination that involves two targets with a ΔpI close to 0 on their own that can cause a flip. Blocking both JAK1 (ΔpI = + 0.05) and JAK3 (ΔpI = 0) (or their complexes with IL4R) can cause a total ΔpI of +0.57 (a synI of 5.71) in the case of certain input combinations (IL-4 combined with either of IL-1 and IL-10).

We also see that the synI of the combinations is generally lower than what we observed in the other direction. Only 38 (11.7%) have a synI over 1, with most (254 or 77.9%) having a synI of 1. This shows that the pathways controlling anti-inflammatory genes are more independent of each other, running parallel in most cases. When viewed together with the above observations about inflammatory signal transduction, it also indicates that the pathways responsible for different aspects of polarization differ not only in the proteins and genes utilized, but their signal transduction follows slightly different logic, resulting in a dissimilar network architecture.

All instances of flips can be found in Appendix A.

## 4. Discussion

Computational models have found widespread use in recent years in many disciplines of biology. They can be used to examine an otherwise unmanageable number of options and possibilities, and they enable us to focus on in vivo and in vitro experiments and efficiently narrow down key targetable molecules. Protein–protein interaction (PPI) and gene regulatory networks (GRN) organize and utilize our knowledge of intracellular events. Due to their scale, they are of particular interest in cancer research and drug target selection.

Modulation of the immune response as antitumor therapy has proven effective, and there is a significant amount of research expanding on the original idea [41,42]. Macrophages have both regulatory and effector functions and exhibit particular responses to the TME; thus, they are valuable targets for such efforts. Moreover, they can adapt to the needs of their particular circumstances via a process called polarization, allowing them to play a role in both the promotion and quenching of inflammation, angiogenesis, tissue repair, and more. The TME is able to exploit this plasticity, pushing macrophages towards states that hinder immune responses and aid tumor survival [43]. Thus, therapeutic strategies aiming to deplete or impede the recruitment of TAMs might negate their undesirable effects. The ideal solution, however, would be to counteract the influence of the TME and drive TAMs towards a state that aids the clearance of cancerous cells.

Here, we constructed an in silico model of the early events in macrophage polarization. Based on PPI databases and then systematically checked and expanded, our model replicates experimentally measured cellular responses to individual stimuli. Using this network, we explored the effects of all possible combinations of our selected inputs and modeled the impact of therapeutic interventions, including mono- and combination therapies. We show that the inhibition of a single target protein has only a moderate effect on the cell’s polarization; in general, this is not capable of reeducating the macrophage and changing an established polarization. However, while still rare, the simultaneous inhibition of two targets can be sufficient, and our model reveals multiple potent combinations with different effect profiles. In most cases, this favorable outcome results from synergistic action on the part of the two simulated therapeutics.

Most methods based on PPI networks use undirected, unsigned interactions, as do the most extensive databases, such as STRING [38] or HPRD [39]. Even though the problem is recognized and systematic solutions are proposed [44], there seems to be limited focus on this essential subject. Our model incorporates data describing both the direction and sign of interactions, and thus, the flow of information in the system, allowing us to characterize interactions to a more profound extent. While our scope is limited to macrophages, our manually curated interaction dataset might be of interest to researchers exploring immunosuppressive processes in the TME. We recommend using the model before in vitro drug testing to accelerate studies and rule in or out molecules that potentially target macrophage polarization in TME.

Our findings may highlight something more fundamental to cellular processes. Examining a different system with a similar method, Fumiã and Martins also found in their cancer model that monotherapies are ineffective in changing cellular states, and that environmental conditions affect the efficacy of treatment strategies [32]. In their system, the accumulation of mutations in cancerous cells is an internal change, while it is external in ours, but in both cases, it might lead to therapeutic interventions losing efficiency over time. Options that show effectiveness in a larger portion of input combinations might mitigate this to some degree and are, thus, of interest.

Models of cellular decisions usually define discrete choices as output nodes, with a limited number of nodes sending edges to them [32,45,46]. In contrast, we aimed to keep as wide a spectrum of output nodes as possible. Here, we present one possible method of evaluation, but adapting it based on different points of view or correcting it based on newly acquired information is easy to execute and does not require a review of the non-output nodes. Researchers interested in a particular gene can explore upstream effects. Supposing the gene in question is not currently included in the model, they can add it if the transcription factors necessary and sufficient for its expression are known without changes to current nodes or edges.

There have been previous computational models investigating macrophage function. The work presented in [47], and [46] based on it, use a similar approach to the one presented here but create much simpler models, with 14 and 21 inner nodes (not in- or output), respectively, compared to our 64. This means that interactions are likely to be indirect and restricts studies to more central nodes, creating an inherent bias. The more recent works seen in [48,49] present comprehensive network structures but restrict themselves to single-node inhibitions. This can mask more subtle effects that do not show notable potency in altering outcomes by themselves and is conceptually farther from actual treatment practices.

Our results highlight the role of STAT6, JAK1 and JAK3, and to a lesser extent Tyk2, STK4 and Sp1, when creating a flip towards a polarization status promoting inflammation and NFAT5 when flipping toward a tissue-protective phenotype. None of these targets are lethal or cytotoxic, according to the Essential Genes database of the International Mouse Phenotyping Consortium (www.mousephenotype.org, accessed 3 January 2022) [50], and thus, could be viable targets for drugs. Frequently administered oncotherapeutic substances including Gefitinib [51] or Imatinib [13] impact STAT6 phosphorylation. The inhibition of JAK1, JAK3, and Tyk2 could partially exert its effect via blocking STAT6 activation in response to IL-4 or IL-13, as evidenced by a blockade of the IL-receptor/JAK complexes with a near identical impact on pI in many cases.

Conversely, the inhibition of various JAKs is used to achieve anti-inflammatory effects. Tofacitinib is used against rheumatoid arthritis and other inflammatory diseases and mainly targets JAK1 and JAK3, with a lesser effect on JAK2 [52]. However, its anti-inflammatory effects are most likely via disturbing the JAK1/STAT1 interaction and not through JAK3 [53]. Moreover, it has been shown to block STAT6 function in fibroblasts [54], and to have anti-angiogenic effects [55], which would be more in line with a shift towards an M1-like state in macrophages. The role of STK4 (also known as MST1) in therapeutic intervention is unclear. Malibatol A inhibits this kinase and promotes an anti-inflammatory shift [56], while treatment with Adapalene activates STK4 and pushes macrophages toward alternative activation [57]. Sp1 is a known target in some forms of cancer, but the effects of such inhibition on TAMs have not yet been explored [58,59,60]. NFAT5 is the target of Arctigenin, a drug shown to drive macrophages towards an anti-inflammatory phenotype in mice [61]. While our results place NFAT5 in a central position when trying to create this shift, in our model, it is only effective as part of a combination of targets. Arctigenin is also reported to affect the phosphorylation status of JAK2 and STAT1 via its primary effect on NFAT5 [61]. Based on their relative position in the signaling network, this is likely to be an autocrine effect or some other kind of feedback, which might be accountable for the difference.

All models necessarily involve simplifications. We focused on the early response phase of polarization and representing its result as part of a continuous spectrum, and did not consider the speed or timeframe of the events. It also means that we focused on protein–protein interactions present in unpolarized cells and excluded pathways or parts thereof that are dependent on changes in gene expression or feedback mechanisms. For example, metabolomic changes are involved in polarization [62] and communication with the TME [63], but they are directed by metabolites such as nitric oxide (NO) that are not produced constitutionally, thus requiring the expression of enzymes (the inducible nitric oxide synthase (iNOS) in the case of NO) to create a signal. Targeting these with outside intervention could have an impact; thus, their inclusion would be a way to improve the model. The Boolean nature of the model also hinders the inclusion of changes in intensity (although some stochastic models could offer an alternative). This limits the model to an output of a static expression pattern instead of a dynamic response and excludes the detailed exploration of effects that would modulate the longevity or stability of the resulting polarization. It also renders the model unable to deal with certain regulatory constructs, e.g., the interplay of IL-6 and IL-10, in which information is encoded in signal duration [64]. The verification of a network this size has other issues. Ideally, the output generated by input combinations should also be considered, but there are not enough data for this to be feasible. Other research groups faced with this problem have also resorted to one-input–one-output verification [65]. A specifically designed high-throughput in vitro test series could provide directly comparable data points, but there might be concerns about the robustness of such methods.

## 5. Conclusions

The current work focuses on the initial intracellular events leading to macrophages changing their phenotype on a continuous spectrum of polarization in response to extracellular stimuli. Here, we presented that the early steps of macrophage polarization, and its results viewed as a spectrum of phenotypes, can be represented with a Boolean network. Moreover, using simulations, we showed that the inhibition of single targets is generally not effective in changing the polarization of macrophages, whereas dual inhibitions can show a synergistic effect and successfully shift macrophage polarization. Our model provides information to accelerate in vitro research and testing, as well as a platform for revealing aspects of macrophage polarization. This is useful for testing immunosuppressive mechanisms and drugs in immunology and cancer research. Further studies are needed to expand our model by including secondary, late-response mechanisms.

## Figures and Tables

**Figure 1 biology-12-00376-f001:**
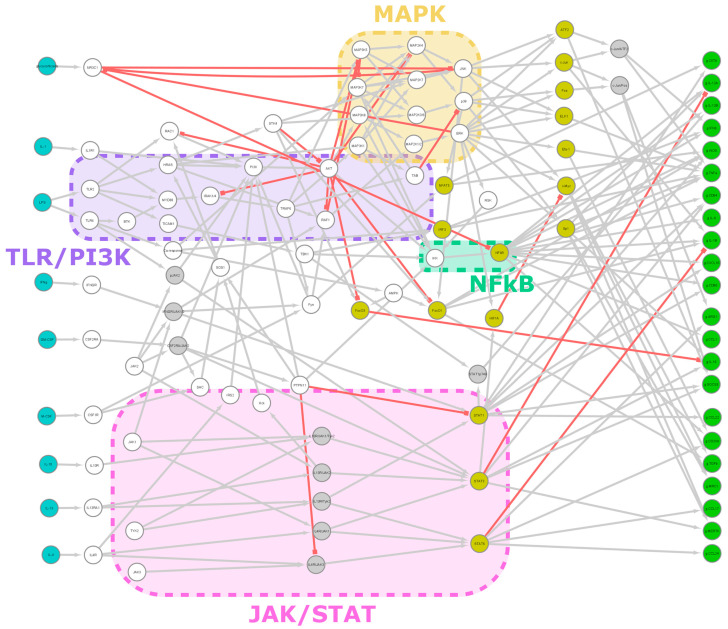
Network overview. A summary image showing all 106 nodes and 217 edges of the model. Input nodes are teal, transcription factors are yellow and output nodes (genes) are green. The grey nodes are complexes or alternative states, represented as separate nodes for technical reasons. Grey edges are stimulatory; red ones are inhibitory. Major signaling pathways are indicated with colored boxes. Edge weight is not indicated here. The base image was created with Cytoscape.

**Figure 2 biology-12-00376-f002:**
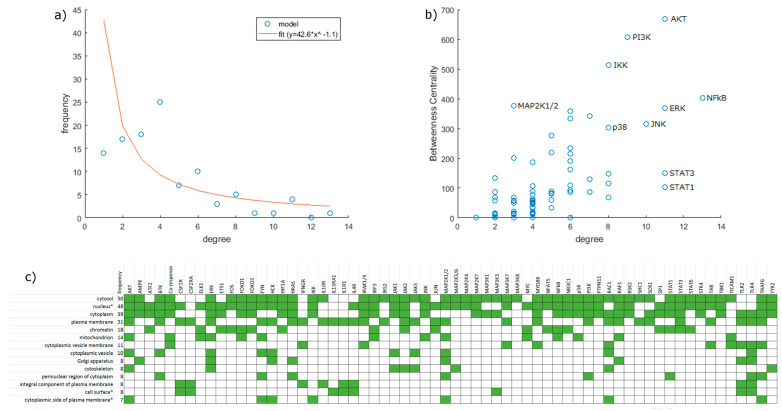
Network statistics. (**a**) Degree distribution of the network. The red line shows a power law curve fitted to the data with degree = 1 and degree = 4 excluded as outliers (R2 = 0.80). (**b**) Betweenness centrality plotted against degree. Some nodes of interest are labeled. (**c**) Cellular localization of proteins represented in the model. Data are based on the Gene Ontology Database. As presented here, terms contain all descendants reachable via “is_a” relations. In specific cases (marked with *), terms with a “part_of” relation were also merged into their ancestors: namely “nucleoplasm” and “nuclear body” were merged into “nucleus”; “external side of plasma membrane” was merged into “cell surface”; and “extrinsic component of cytoplasmic side of plasma membrane” was merged into “cytoplasmic side of plasma membrane”.

**Figure 3 biology-12-00376-f003:**
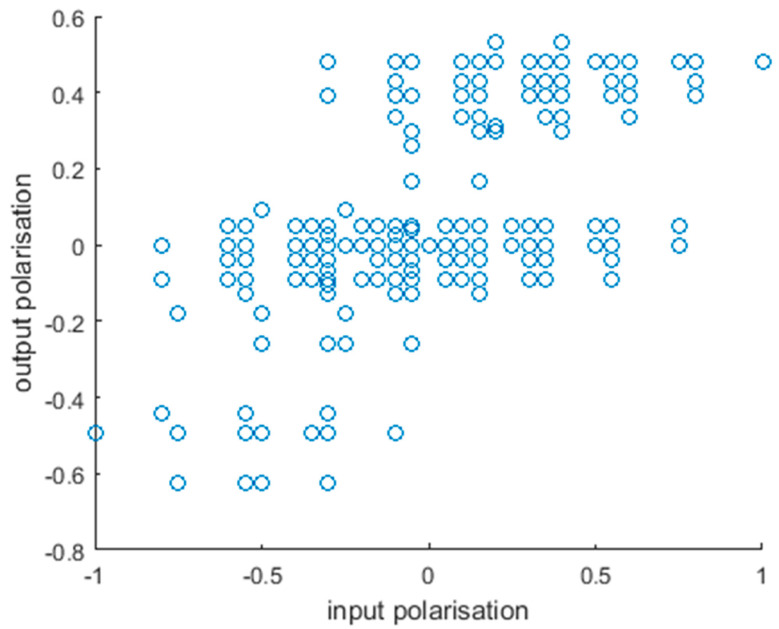
Alignment of input and output polarization. Each point represents an input combination. While input polarization does not translate directly to the output, there is a clear correlation between the two, further supporting that our model is in line with our understanding of the effects of these factors.

**Figure 4 biology-12-00376-f004:**
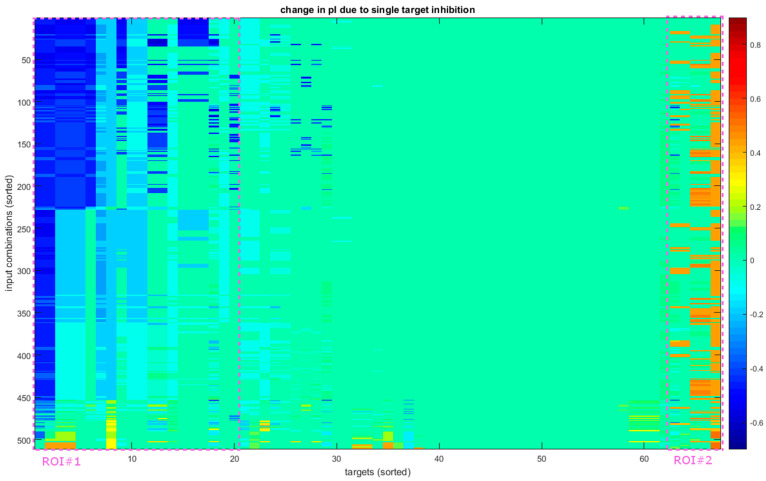
Reactions of the system to single-target inhibition. The vertical axis shows all 511 input combinations and the horizontal all 67 perturbed nodes (direct inhibition of input and output nodes is not performed, extending to receptors with the exception of TLR2 and 4). Both axes are sorted for presentation purposes based on the first principal component of the data. The highlighted areas show the portion of instances that shift predominantly towards a regenerative function (ROI#1) or a more inflammatory one (ROI#2). These only cover approximately 30% and 7% of targets, respectively, and are not homogenous, showing that most inhibitions have a limited effect on the state of the cell.

**Figure 5 biology-12-00376-f005:**
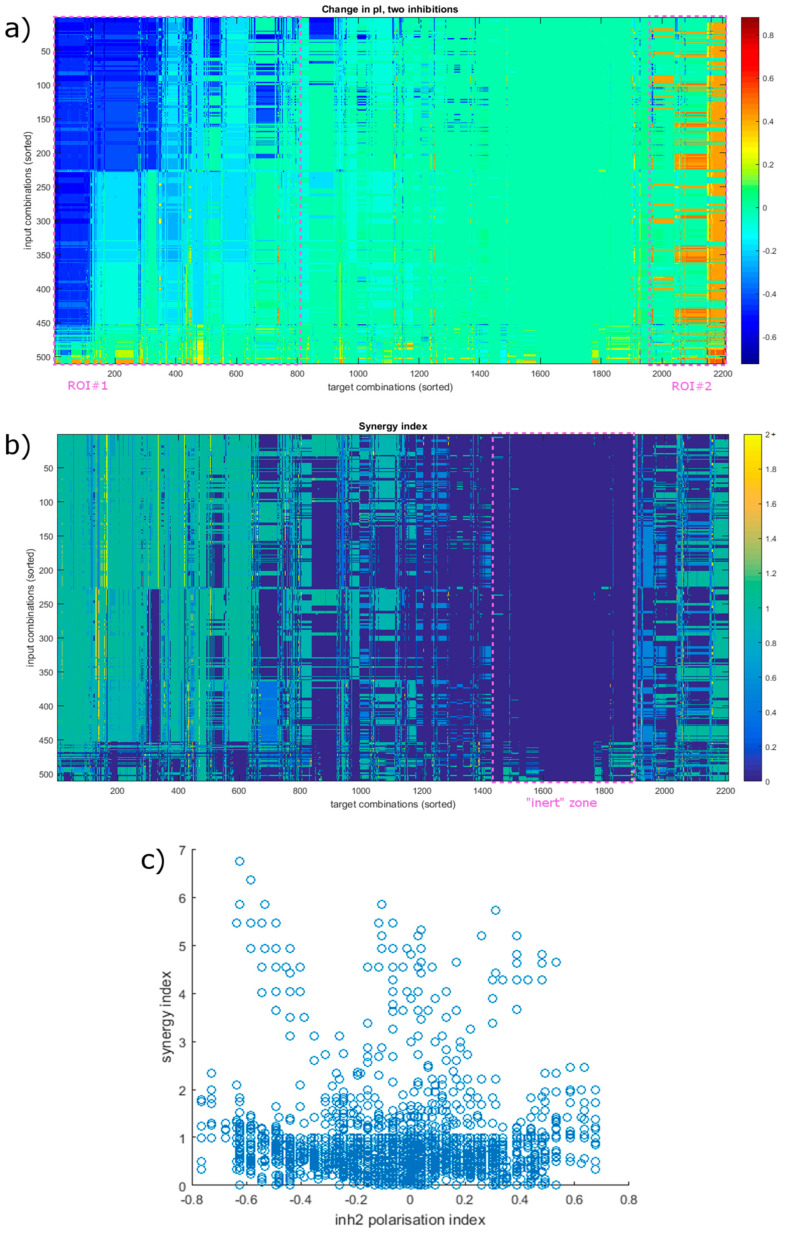
Effects of the simultaneous inhibition of 2 nodes. (**a**) Changes in the polarization index in response to combined inhibition. Areas that shift predominantly towards one state are highlighted (as ROI#1 and #2), similar to Figure 4. They are slightly larger, with approximately 37% of target combinations mainly causing a shift in a regenerative direction and 11% towards an inflammatory state (compared to 30 and 7% for individual target inhibition, see Figure 4.) (**b**) Synergy index in all conditions. The ROIs present in previous heatmaps are not clearly visible, except the mostly inert zone (marked) where the inhibition has little to no effect. High synergy spots appear scattered compared to the ΔpI. The axes in (**a**,**b**) are sorted for presentation purposes, and the order is the same in both, based on ΔpI. (**c**) Synergy index is achieved in relation to a shift in the polarization index. Values are somewhat cropped due to the 𝜏 adjustment parameter (see Equation (3) in Materials and Methods) capping synI at ΔpI/𝜏.

## Data Availability

Network node and edge data, as well as the results of simulations and MATLAB code for the Boolean model, are all accessible in the electronic Appendix A.

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
