# Peer review of "Effective Reversal of Macrophage Polarization by Inhibitory Combinations Predicted by a Boolean Protein–Protein Interaction Model"

_biology, 2023, doi:10.3390/biology12030376_

Round 1

Reviewer 1 Report

Dear authors:  As a reviewer with intense interest in this topic, I appreciate the effort you put into this paper.   I do understand how and why you focused the model(s) on the subject of macrophage polarization, but I take issue with the notion that macrophages are either M1 or M2....this is not the case.  Macrophages are in all sorts of physiological states depending on their specific tissue milieux.   Often these differences are determined by energetics, and how committed macrophages are full OXSPHOS vs aerobic glycolysis, and how committed they may be to one or the other.  Along these lines of thought, you did not mention the IL-10 influence on polarization via controlling metabolism (mTOR involvement).

Iam not one of the authors of the following citations, but I am familiar with the work coming out of NIH in the USA within the context of immunometabolism.   Before I list these citations, I wish to comment on the model.   What other models are available, what are their pros and cons, and why did you choose this particular model.   Again, nice work.... but I think your paper will be much better with these above comments.

Nat Commun. 2020; 11: 698.

Published online 2020 Feb 4. doi: 10.1038/s41467-020-14433-7

Metabolites. 2020 Nov; 10(11): 429.

Published online 2020 Oct 26. doi: 10.3390/metabo10110429

Br J Pharmacol. 2019 Jan; 176(2): 155–176.

Published online 2018 Nov 6. doi: 10.1111/bph.14488

J Clin Invest. 2018 Aug 31; 128(9): 3794–3805.

Published online 2018 Jul 30. doi: 10.1172/JCI99169

Nat Commun. 2017; 8: 2074.

Published online 2017 Dec 12. doi: 10.1038/s41467-017-02092-0

Sincerely

Author Response

Response to Reviewer #1

Comment: I do understand how and why you focused the model(s) on the subject of macrophage polarization, but I take issue with the notion that macrophages are either M1 or M2....this is not the case.  Macrophages are in all sorts of physiological states depending on their specific tissue milieux.

Answer: We thank Reviewer #1 for the observation. In our study, we aimed to depict macrophage polarization as a continuous spectrum. 

Accordingly, we clarified this throughout the manuscript and emphasized the continuous nature of the spectrum. We also cited the spectrum model of polarization, and also our own polarization index has a continuous value set.

There is only one case where we had to introduce a more granular approach, in the definition of flips (repolarization in response to perturbation). A simple difference in our continuous polarization index would not have sufficed there, as a change from 0.8 to 1 is markedly different in effect from another taking the cell from -0.2 to +0.2. In order to give a clear picture of which perturbations (inhibition of a target) would be of interest in further studies, we had to define threshold values. We have revised the text in Methods/Verification referencing Fig.3. to clarify that this is the purpose of the thresholds. Since the raw output of the model is an expression pattern with over 130 million theoretically possible states, this is not inherent to the model, and would not be necessary for all questions examined with the network.

Comment: Often these differences are determined by energetics, and how committed macrophages are full OXSPHOS vs aerobic glycolysis, and how committed they may be to one or the other.  Along these lines of thought, you did not mention the IL-10 influence on polarization via controlling metabolism (mTOR involvement). [The reviewer has also recommended articles about immunometabolism.]

Answer: We thank Reviewer #1 for pointing out the importance of metabolic processes in macrophage polarization. Our aim was to characterize the first steps that are responsible for the primary decision of which way to polarize. Thus feedback effects (like auto/paracrine events) and steps dependent on changes in gene expression were not included in our study. Accordingly, we mention in the Discussion as one the limitations of the model that it does not encompass the entirety of processes that lead to a fully polarized cell. This is one of the ways the model could be improved and refined in the future.

Also, we tried to include TOR complexes, but their direct interactions with other elements were not described with enough clarity in the literature for us to be able to translate them into the model sufficiently. We could mostly find indirect effects, and we do not include such in our model (except for not making PIP3 a separate node).

I would also note that NOS2/iNOS (an enzyme responsible for producing NO, the subject of 3 of the linked articles) is an output in our model and its activation is taken into consideration when calculating the polarization index, even though the why and how of its contribution is not explored. However, we added a section about this topic in the Discussion section and included from the recommended citations.

Reviewer 2 Report

The study, titled "Effective Reversal of Macrophage Polarization by Inhibitory Combinations Predicted by a Boolean Protein – Protein Interaction Model" developed a Boolean control network model to evaluate the intracellular processes leading to macrophage polarization in response to the tumor microenvironment. The work is interesting, but since the authors developed an in silico model, I think the model should be based on strong assumptions, i.e., they should be careful in data curation. Therefore, the authors need to improve the manuscript in the following points;

1.       In the manuscript it was stated that “The input (receptors/ligands) and output nodes (genes) were selected based on our current understanding of macrophage polarization” which is not clear. The authors should briefly mention about their selection criteria.

2.       In the manuscript it was stated that “Some nodes were added based on expert decision during the curation process”, which is not clear and scientific. What do the authors mean by the term "expert decision"?

3.       Why were the MAPK, TLR, and Jak/ STAT paths selected to include all input and output nodes?

4.       CIBERSORT is a deconvolutional algorithm that uses the genes to predict the proportions of 22 immune cell types (including T cells and macrophages). This source includes a large amount of data. I recommend that the authors use these data in the construction of the network.

5.       Interactions are from various sources, including STRING. Which version was used in the study? Because it was written that the data was accessed in 2021, which is quite old. The database STRING is constantly updating the data it contains. In order not to miss any information, I recommend researchers to consider the latest data.

6.       BIOGRID is an open-access database resource that contains protein interactions, and it contains human-specific interactions that are supported in much greater numbers by literature evidence. Why did the authors not consider these data in the data curation?

7.       Macrophage-related genes have been identified in the human genome. A total of 27 M1- and M2-type macrophage proteins were included in the study. The authors believe that this number is sufficient to evaluate, present, and verify their model?

8.       It was known that macrophages can polarize into two distinct states (M1 and M2) with different phenotypes and functions. For this reason, the manuscript states, one can predict that "M1- and M2- like signaling has different network architecture " So I do not think this is a new result or conclusion.

9.       It should be clearly stated in the manuscript that if other researchers use this model in their study, what will they find out?

10.   Some of the figures were not legible (especially Figure 1 and Figure 2).

Author Response

Response to Reviewer #2

Thank you for reading the manuscript thoroughly and taking the time to comment in detail.

Comment #1: In the manuscript it was stated that “The input (receptors/ligands) and output nodes (genes) were selected based on our current understanding of macrophage polarization” which is not clear. The authors should briefly mention about their selection criteria.

Answer: We thank the Reviewer for the comment. One of our primary concerns was to create a model that can be verified based on existing research data. In addition, due to the manual curation process necessary we were already restricted to a model more limited in size. Also, common sources of high throughput data do not contain parameters that could be used as inclusion criteria (representing the importance of a given entity in this given process). Thus we have decided to carry out a systemic literature search based on macrophage polarization, but with the elimination of articles where raw data/source was not available and direct interaction between the studied proteins were not proven experimentally.

We have included clarification in the manuscript, in the Methods section.

Comment #2: In the manuscript it was stated that “Some nodes were added based on expert decision during the curation process”, which is not clear and scientific. What do the authors mean by the term "expert decision"?

Answer: We thank the Reviewer for the correction. After the systematic search for elements of the system, some verification criteria remained unsatisfied. We hypothesized that an element missing from the model is responsible for the discrepancy and have performed additional directed manual searching to identify such missing nodes, if that was indeed the case. During this process nodes not present or sufficiently connected in the databases were included in the model.

We have included clarification regarding this issue in the resubmitted text, in the Methods section.

Comment #3: Why were the MAPK, TLR, and Jak/ STAT paths selected to include all input and output nodes?

Answer: We thank the Reviewer for pointing out the need for clarification here. In order to translate the processes observed in the actual cell into our framework with Boolean logic, a considerable amount of information on the underlying events is necessary, not only describing the chemical events, but also their effect on the behavior of the entity in question (e.g. most STAT family proteins can be phosphorylated on multiple amino acids with differing consequences). We have observed that even in the case of proteins known and studied for decades there can be quite a bit of uncertainty, creating an obstacle when constructing the model. Thus we have concentrated our efforts on the pathways with the highest amount of information backing them up and selected enough of them to include all the input and output nodes.

We have included clarifications in the Methods, Nodes section in the revised manuscript.

Comment #4: CIBERSORT is a deconvolutional algorithm that uses the genes to predict the proportions of 22 immune cell types (including T cells and macrophages). This source includes a large amount of data. I recommend that the authors use these data in the construction of the network.

Answer: We thank the Reviewer for bringing this algorithm to our attention. Genes are picked in CIBERSORT based on their ability to discriminate between all 22 types of cells they have considered. They are also incentivized to minimize the number of genes their procedure is using. Making it a mostly artificial selection of genes, chosen for a particular purpose. Out of the 27 outputs of our model they only consider 7 (plus 1 indirectly, CIITA is a major regulator of HLA genes). In all of these cases, our decision to flag the gene leaning toward M1 or M2 is in accordance with the data used by CIBERSORT.

By a quick estimate, they have about an additional 160 genes that show some level of differential expression in different macrophage sub-populations. Since no comprehensive database of transcription factor effects is available, attempting to include these in the model would mean checking all of them manually for any and all TF-s controlling them. Since the database is not meant to provide any sort of measure of impact on cell phenotype, the cost-benefit ratio of such an endeavor would be prohibitively high. (If we were to acquire and use their complete microarray data, it would only make this less efficient.)

CIBERSORT is designed to analyze mixtures of cells, while ours is a single-cell model. On their homepage they note that “Clustering and de novo identification of cell types from single cell RNA sequencing data is not supported at this time.”. Nevertheless, we have attempted to compare our polarization index to a decision made with CIBERSORT. We have entered our binary outputs from unperturbed runs of the model as “mixtures”, having expression levels of 0 and 100. When the signature matrix is the full LM22 dataset, only 2 of 13 registers considerable (20%+) levels of polarized macrophages, with 5 more reporting some monocytes. This is insufficient for a comparison. Restricting the LM22 set to only macrophages returns an error. Including Monocytes and all 3 forms of macrophages does produce a result. We still do not get basically any M1 presence detected, most cases reporting 89%+ of monocytes or M2 macrophages. If we consider any non-M2 results as M1, 11 of 13 cases are in line with our polarization index (See figure below).

Unfortunately, this validation step was only successful in the case of the M2 side of the spectrum, due to technical reasons and the incompatibility of the CIBERSORT system with our PPI model. Thus, we hope that the expert Reviewer would understand that we could not include this incomplete finding in our manuscript. Although, in the next version of our model and if/when CYBERSORT enables single-cell features as well, it would serve as a useful validation method for simulation outcomes.

Comment #5: Interactions are from various sources, including STRING. Which version was used in the study? Because it was written that the data was accessed in 2021, which is quite old. The database STRING is constantly updating the data it contains. In order not to miss any information, I recommend researchers to consider the latest data.

Answer: We thank the reviewer for pointing this out. A large portion of the model consists of well-researched proteins and pathways that have been known for decades. Furthermore, the finalization of edges, that define the functionality of the model were not determined by STRING, whose data on interactions would not have been suitable for many interactions and/or would have failed our inclusion criteria. The STRING database only provided a framework or pool for interactions (edges) in our model that were later meticulously filtered and curated based on our criteria.

Comment #6: BIOGRID is an open-access database resource that contains protein interactions, and it contains human-specific interactions that are supported in much greater numbers by literature evidence. Why did the authors not consider these data in the data curation?

Answer: We primarily relied on HPRD for reference articles, and to a lesser extent SignaLink. This was sufficient for most of the interactions, and we could acquire the remainder with manual searches, mainly using the PubMed database. In a few cases requiring manual search we have made queries in BioGrid, but did not find reference data in addition to what was present in HPRD, and thus decided not to cite it as source.

Comment #7: Macrophage-related genes have been identified in the human genome. A total of 27 M1- and M2-type macrophage proteins were included in the study. The authors believe that this number is sufficient to evaluate, present, and verify their model?

Answer: We thank the Reviewer for the question. Our primary goal when designing the output was to detect the decision of the cell regarding polarization. It was not our intention to give a comprehensive gene expression signature comparable to RNAseq or microarray results. The number of output points in our model is limited partially by the constraints we imposed on the system, and partially by necessity. The former source of simplification is our decision to limit our model to the early stages of polarization and exclude feedback routes and events dependent on changes in gene expression. We have also considered other genes for inclusion, but the scarcity of data about transcription factor binding and gene expression has hindered us greatly. For example, the product of the VEGFA gene undergoes post-transcriptional changes that significantly alter its effect in the context of the TME. The molecular entities responsible for these events are mostly unknown, however. Without including them, the implications of an active VEGFA gene would be ambiguous, thus we had to abandon it.

Still, we were able to incorporate numerous different combinations and interactions of transcription factors, with only 1 pair of genes having the same incoming set of edges (CCL17 and CCL24). The output genes are not considerably numerically biased toward any type of polarization (we also correct for any such effect in the way we construct the polarization index). All output genes were previously observed to change expression during polarization (see the references in Supplementary Table 3).

Other similar studies have also been presented with output points in the same order of magnitude as those utilized by us (like [Liu 2021] with 39 genes). Our output also greatly exceeds the number of output points employed by earlier models (like [Marku 2020] that has 3 and [Palma 2018] that has 2).

Comment #8: It was known that macrophages can polarize into two distinct states (M1 and M2) with different phenotypes and functions. For this reason, the manuscript states, one can predict that "M1- and M2- like signaling has different network architecture " So I do not think this is a new result or conclusion.

Answer: We thank the Reviewer for pointing out the lack of clarity here. Our observation was made specifically about the underlying network structure and its construction. We did not expect a difference beforehand, as the logic of signal transduction can be greatly similar in pathways responsible for very distinct functions. We have added clarification in the Results section and removed the statement from the Conclusions in order to focus the message of the article elsewhere.

Comment #9: It should be clearly stated in the manuscript that if other researchers use this model in their study, what will they find out?

Answer: We agree with the Reviewer, and thank them for their insight. We have included text clarifying this issue in the Discussion and Conclusions sections to remedy our previous oversight.

The primary intended use of our model is as a preliminary step in preparation of in vitro studies in search of bioactive molecules to be used in immunotherapies targeting macrophages. By simulating the inhibition of proteins, the expected impact of drug candidates on macrophage polarization can be assessed and the results can be used as a filter when deciding the subjects of further, more costly steps of development. It can also be utilized by researchers interested in the expression of genes in this particular context.

Comment #10: Some of the figures were not legible (especially Figure 1 and Figure 2).

Answer: Loss of resolution is probably due to PDF-conversion. We have provided high resolution images with our submission, readers should be able to zoom in if desired. We guarantee that in the final version of the manuscript, all Figures will be displayed at maximum resolution.

Round 2

Reviewer 1 Report

This is a very good paper, and the authors clearly responded to my concerns.  This is an improved body of work that will be very important to those working in the field.

Reviewer 2 Report

During the revision period, I think authors addressed all the comments and suggestions. Therefore, I believe that the manuscript is more valuable in the current form.